# The Impact of the Coronavirus Pandemic on the Contribution of Local Green Space and Nature Connection to Mental Health

**DOI:** 10.3390/ijerph20065083

**Published:** 2023-03-14

**Authors:** Claire L. Wicks, Jo L. Barton, Leanne Andrews, Sheina Orbell, Gavin Sandercock, Carly J. Wood

**Affiliations:** 1School of Health and Social Care, University of Essex, Colchester CO4 3SQ, UK; 2School of Sport, Rehabilitation and Exercise Science, University of Essex, Colchester CO4 3SQ, UK; 3Department of Psychology, University of Essex, Colchester CO4 3SQ, UK

**Keywords:** depression, anxiety, stress, well-being, natural environment

## Abstract

Background: Exposure to green space and feeling connected to the natural environment have independently been associated with improved mental health outcomes. During the coronavirus pandemic, people experienced restrictions on access to the outdoors, and health data indicated a decline in mental health in the UK general population. Methods: Data available from two independent surveys conducted prior to and during the pandemic enabled a naturally occurring comparison of mental health and its correlates prior to and during the pandemic. Results: Survey responses from 877 UK residents were included in the analyses. Independent *t*-tests revealed significant declines in mental health scores during the pandemic. After controlling for age and gender, greater nature connection significantly predicted lower depression and stress and improved well-being. Percentage of green space did not significantly predict any mental health outcomes. Further, time point (pre- or during COVID) and the interaction of time point with green space and nature connection did not significantly predict any of the outcome measures. The findings indicate that nature connection may play an important role in promoting mental health. Strategies to improve mental health and reduce mental illness should consider the role of nature connection and the use of interventions that involve direct interaction with natural environments.

## 1. Introduction

Mental health is defined as “a state of well-being in which every individual realises their own potential, can cope with the normal stresses of life, can work productively and fruitfully, and is able to make a contribution to her or his community” [1]. By contrast, a mental illness is characterized by a clinically significant disturbance in an individual’s cognition, emotional regulation, or behaviour that is associated with distress or impairment in important areas of functioning, such as work, daily activities, or personal relationships [2]. Globally, in 2019 one in eight people were living with a mental illness [2], whilst in the UK one in six individuals experience a common mental illness such as depression at any time [3]. Poor mental health is one of the main causes of disease burden worldwide [4] and cost the UK economy an estimated GBP 117.9 billion annually [5].

Evidence suggests that the coronavirus pandemic resulted in further declines in mental health. Throughout the pandemic, mental health declined in the general UK population [6,7], along with increased rates of loneliness, social isolation, and mental illness [7,8,9]. In the first year of the pandemic, depression and anxiety were reported to have increased by more than 25% [10]. A UK-based study reported a near-50% increase in the prevalence of depression since the onset of the pandemic [11]. The negative mental health effects of the pandemic may also be heightened in disadvantaged populations, further increasing health inequalities [12,13]. For example, younger adults, women, and those from minority ethnic groups experienced greater negative mental health impacts during the pandemic [13], whilst individuals with existing mental illness were more vulnerable to declines in mental health due to factors such as disruptions to mental health services [13,14,15,16].

Evidence of the benefits of exposure to green space for mental health is growing [17]. Green space is broadly defined as any vegetated land or water, often found within an urban area, including parks, gardens, allotments, playing fields, grassed areas, rivers, and canals. It also includes other natural areas, such as woodlands, forests, and wilderness areas. Evidence suggests that green space close to the home, including urban green space, is associated with a reduced risk of developing anxiety and depression, more positive emotions, greater life satisfaction and reduced loneliness [18,19,20,21,22,23,24,25]. Factors proposed to mediate the relationship between green space and mental well-being include air quality and opportunities for physical activity; however, evidence about the factors mediating the relationship between green space and mental illness are less clear [26]. Furthermore, evidence has suggested socioeconomic, and demographic factors such as age and gender may moderate the green space–health relationship [27]. These variables should therefore be accounted for in analyses examining associations between green space and health.

Nature connectedness, defined as the degree to which an individual feels connected to the natural world [28], is also associated with positive mental health outcomes, including improved well-being, lower likelihood of medication for depression, reduced mental distress, increased happiness, and higher life satisfaction [29,30,31]. Evidence suggests that nature contact may also mediate the relationship between nature connectedness and health, as individuals who are more connected to nature are also more likely to visit green spaces and thus derive the associated health benefits [32]. 

The impact of the pandemic on use of green space is unclear. In line with UK government coronavirus restrictions, after 23 March 2020, individuals could only leave their homes for the following reasons: (i) to shop for necessities, for example, food and medicine; (ii) for one form of exercise a day, alone or with household members; (iii) for any medical need or to provide care or to help a vulnerable person; and (iv) to travel for work purposes if the work could not be conducted from home. Whilst some studies reported increased use of green space during this time [33], other studies reported reductions in time spent visiting green space [34]. Those from lower socio-economic groups were also reported to be less likely to visit green space, with increases in inequality of green space access further disadvantaging the most vulnerable groups in society [33,34]. Despite mixed evidence regarding use of green space during the pandemic, exposure to green space was associated with fewer symptoms of anxiety, whilst the quality of views of green space from home was associated with improved well-being [35,36].

To date, the effect of the pandemic on the potential mental health benefits derived from nature connection and local green space has not been investigated. The aims of this study were therefore to: (i) examine changes in mental health outcomes from before to during the pandemic; (ii) confirm the relationship between percentage of local green space and nature connection with mental health outcomes; and (iii) explore whether the relationship between exposure to green space and nature connection with mental health outcomes differed before and during the coronavirus pandemic.

## 2. Materials and Methods

### 2.1. Participants

Participants were recruited separately to take part in two independent online surveys, both of which included questions about nature connection and mental health. Studies were advertised via social media, including Facebook and Twitter, using a snowballing sampling strategy. Details of the studies were shared amongst the researchers’ social media groups and on institutional and research group websites. The researchers also shared details of the studies with their colleagues, collaborators, and contacts, all of whom were asked to share the survey with their networks. All participants were aged ≥18 years and residing in the UK at the time of survey completion.

### 2.2. Procedures

The two surveys were completed between 28 August 2019 and 2 June 2022, with study one taking place between 28 August 2019 and 30 April 2020 and study two taking place between 1 May and 2 June 2020. Study one was undertaken as part of a PhD study exploring the relationship between physical activity outside in nature and mental health and well-being outcomes. Study two was undertaken to explore changes in physical activity and well-being during the UK coronavirus lockdown. All participants that completed either survey on or after 23 March 2020 did so during the UK government’s coronavirus restrictions. For the purposes of the present analyses, participants completing a survey before 23 March were allocated to the pre-pandemic group and those completing a survey on or after 23 March were allocated to the during-pandemic group.

All participants completed the surveys electronically via Qualtrics and provided their consent to take part in the study after reading the participant information sheet on the landing page of each survey. In both surveys, participants were asked to provide demographic information and to complete measures assessing their well-being, depression, anxiety and stress, and nature connection. Ethical approval was granted by the School of Health and Social Care and the School of Sport, Rehabilitation and Exercise Sciences Ethics subcommittees at the University of Essex (Ref: ETH2122-0071 & ETH1920-1283).

### 2.3. Measures

The following measures were used for both datasets.

#### 2.3.1. Sociodemographic Measures

Participants were asked to provide a range of demographic data, including age, gender, ethnicity, employment status, relationship status, and postcode.

#### 2.3.2. Warwick Edinburgh Mental Well-Being Scale Short Form

Participants’ well-being in the last month was assessed via the short form Warwick Edinburgh Mental Well-Being Scale (SWEMWBS) [37,38]. The SWEMWBS consists of seven positively worded items from the full 14-item scale, e.g., “I’ve been feeling optimistic about the future.” The well-being score is calculated by summing responses to each item, which are scored on a five-point Likert scale scored from 1 (none of the time) to 5 (all of the time). The raw SWEMWBS scores were converted to metric scores [39] prior to analyses to produce scores ranging from 7 to 35, with a higher score indicating a higher level of well-being. The SWEMWBS has been reported to have a Cronbach alpha of 0.84 using England population-level data [37], with a correlation between the full and short versions of *r* = 0.954 [39]. In the current sample, Cronbach’s alpha was 0.91, 0.83 and 0.90 for the three scales, respectively, indicating very good reliability. The UK normative well-being score pre-pandemic was 23.7 ± 3.9 [37]. During the pandemic, a mean score of 20.8 ± 5.1 was also reported by Smith et al. [5].

#### 2.3.3. Depression, Anxiety and Stress Scale

The Depression, Anxiety and Stress Scale 21 is a 21-item scale that is designed to measure the emotional states of depression, anxiety and stress [40]. Each of these sub-scales contains seven items, e.g., “I felt that I had nothing to look forward to,” with response categories from 0 (did not apply to me at all) to 3 (applied to me very much or most of the time). The overall score for each sub-scale is calculated by summing the items and multiplying by two, with scores ranging from 0–42 and a higher score representing greater feelings of depression, anxiety or stress. The sub-scales have previously been demonstrated to have reliabilities of 0.88, 0.82 and 0.90 respectively [40]. In the current sample the sub-scales had Cronbach alphas of 0.91, 0.83 and 0.90, respectively, indicating very good reliability. The UK normative values for depression, anxiety and stress are 5.66 ± 7.74, 3.76 ± 5.90, and 9.46 ± 8.40, respectively [40].

#### 2.3.4. Nature Connection Index

The extent to which participants felt connected to the natural environment was assessed via the Nature Connection Index (NCI) [41]. The NCI consists of six items, e.g., “Spending time in nature is very important to me,” with each item scored on a seven-point Likert scale from 1 (completely disagree) to 7 (completely agree). The raw scores are converted using a weighted points index and the converted scores summed to calculate the overall NCI score. Scores range from 0 to 100, with higher scores indicating higher connectedness to nature. The NCI correlates highly with other measures of nature connectedness, including the Nature Relatedness Scale short form (*r* = 0.67), which is frequently used in nature and green exercise research [41]. The measure has been developed and tested in the UK general population and is reported to have a Cronbach alpha of 0.92. In the current sample, the Cronbach alpha was 0.87, indicating very good reliability. The UK normative nature connection score for adults is 61.16 ± 27.88 [41].

#### 2.3.5. Percentage Green Space in Ward

Participant’s postcode data were used to identify their census area statistic ward (2001), which was subsequently used to determine the percentage of green space within each participant’s ward using the data of Richardson and Mitchell [42]. This percentage value was used as an indicator of local green space.

### 2.4. Statistical Analysis

Incomplete responses were removed from the datasets prior to analysis (*n* = 87.9%). SPSS (V.28) was used for analysis with significance set at a *p* value of less than 0.05. Since the data were derived from independent non-equivalent samples, preliminary analyses were conducted to compare the two samples. Independent *t*-tests were used to compare age, nature connection, and percentage green space, whilst a chi-squared test of independence compared the distribution of sex, ethnicity, employment, and relationship status in the two samples.

To examine changes in mental health outcomes from before to during the coronavirus pandemic (aim i), independent *t*-tests were used to compare depression, anxiety, stress, and well-being between the participants who completed the survey at the two time points. One-sample *t*-tests were also used to compare scores to the UK normative values for each of the measures.

Generalized linear modelling was conducted to explore multivariate predictors of depression, anxiety, stress, and well-being. For depression, anxiety, and stress, we used generalized linear models with Tweedie log-link function and employed robust estimates of covariance due to the skewness of the data and inclusion of zero responses. For well-being, we used a linear function and employed robust estimates of covariance. In Model 1, age and gender were entered as predictors to adjust for the effect of these demographic variables on outcome measures. Rather than testing the effect of multiple demographic variables in this study, we only included age and gender, as they have previously been reported to influence the green space–health relationship [27]. In Model 2, nature connection and the ward’s percentage of green space were added as predictors (aim ii). In Model 3, study time point (pre- or during COVID) was added as a further predictor, with the main effect (aim i) and interaction effect (aim iii) with nature connection and green space percentage on the outcome variables being explored. The fit of the models was examined using the Akaike information criterion (AIC), with a lower value indicating better model fit. Each model was compared to the intercept-only model using a likelihood ratio chi-squared test. The Wald chi-squared test was used to examine the strength of each predictor variable after controlling for the predictor variables already entered into the model.

## 3. Results

### 3.1. Participants

Table 1 displays demographic details of participants. Overall, 877 participants completed the surveys, with 470 doing so pre-pandemic and 407 on or after 23 March 2020. All participants completing the survey pre-pandemic were from study one (*n* = 470), whilst participants completing the study during the pandemic were from both study one and study two (study one *n* = 92; study two *n* = 315). An independent *t*-test revealed that the pre-pandemic sample were significantly older than the during-pandemic sample (*t*(873) = 7.87; *p* < 0.001), whilst a chi-squared test for independence revealed that there was a significantly smaller proportion of females in the pre-pandemic sample (χ^2^ = 18.51; *p* < 0.001). A chi-squared test of independence also revealed significant differences between the pre-pandemic and during-pandemic sample in terms of employment status (χ^2^ = 19.13; *p* < 0.001), with a greater percentage of the during COVID sample being in full-time employed. Overall, a majority of participants were of a white background and in full-time employment both before and during the pandemic (Table 1). A majority of participants were also married.

Independent *t*-tests revealed a significantly lower percentage of green space within the ward of participants who completed the survey during the pandemic compared to pre-pandemic (*t*(619) = 2.949; *p* = 0.003). There were no significant differences in nature connection scores between participants who completed the survey before and during the pandemic (*p* > 0.05). Nature connection scores were significantly higher than the reported normative values by Richardson et al. [41] for both before (*t*(469) = 7.209; *p* < 0.001) and during (*t*(406) = 7.037; *p* < 0.001) the pandemic and for the total sample (*t*(876) = 11.16; *p* < 0.001).

### 3.2. Univariate Comparisons before and during the Pandemic

Independent *t*-tests revealed significant differences between participants who completed the survey before and during the pandemic for depression (*t*(875) = −4.061; *p* < 0.001), anxiety (*t*(817) = −4.528; *p* < 0.001), stress (*t*(875) = −6.126; *p* < 0.001) and well-being (*t*(875) = 2.896; *p* = 0.004). Participants who completed the survey during the pandemic had significantly worse scores on all well-being variables (Table 2).

One-sample *t*-tests also revealed significantly lower well-being scores than the normative value of 23.7 ± 3.9 reported by Fat et al. [37] both pre (*t*(469) = −6.403; *p* < 0.001) and during (*t*(406) = −10.319; *p* < 0.001) the pandemic. The pandemic mean was significantly higher (*t*(406) = 4.833; *p* < 0.001) than the mean pandemic score of 20.8 ± 5.1 reported by Smith et al. [5]. Depression scores were significantly higher than the normative value of 5.66 ± 7.74 [40] both pre (*t*(469) = 4.537; *p* < 0.001) and during the pandemic (*t*(407) = 10.151; *p* < 0.001). Neither anxiety or stress scores pre-pandemic were significantly different (*p* > 0.05) from the normative values of 3.76 ± 5.90 and 9.46 ± 8.40, respectively. During the pandemic, both anxiety (*t*(405) = 5.606; *p* < 0.001) and stress (*t*(407) = 7.934; *p* < 0.001) were significantly higher than the normative values of Henry and Crawford [40]. However, during the pandemic, anxiety and stress scores still fell within the range for ‘normal’ severity as per the published DASS-21 cut points, whilst depression was on the border of the ‘normal’ and ‘mild’ categories.

### 3.3. Generalized Linear Models

#### 3.3.1. Depression, Anxiety and Stress

All models were significantly different from the intercept model, with Model 3 providing the optimal model fit for depression, anxiety, and stress according to AIC (Table 3). Age was a significant predictor of depression, anxiety, and stress in Model 1 (Table 4), with older age predicting lower depression, anxiety, and stress. Gender was also a significant predictor of both anxiety and stress in Model 1, where being female predicted greater anxiety and stress.

Age remained a significant predictor of depression, anxiety, and stress in models two and three; however, gender was no longer a significant predictor of anxiety or stress after including nature connection and ward percentage of green space in the model (Table 4). Higher nature connection was significantly associated with lower depression and stress in models two and three. In Model 3, neither the time point of study completion or the interaction of time point with nature connection and green space percentage significantly predicted depression, anxiety, or stress. For depression, the interaction of time point and nature connection was approaching significance (*p* = 0.057) such that there was a stronger association of higher nature connection with lower depression pre-COVID compared to during the coronavirus pandemic.

#### 3.3.2. Well-Being

All models yielded a significant overall model effect, with Model 3 providing the best model fit (Table 3). Age was a significant predictor of well-being in Model 1 and 2 (Table 4), with older age being associated with higher well-being. In Model 2, higher nature connection was also associated with better well-being. In Model 3, neither time point of study completion or the interaction of time point with nature connection and green space percentage significantly predicted well-being. Nature connection remained a significant predictor, whilst the effect of age became non-significant (Table 4).

## 4. Discussion

The aims of this study were to: (i) examine changes in mental health outcomes from before to during the pandemic; (ii) confirm the relationship between percentage of local green space and nature connection with mental health outcomes; and (iii) explore whether the relationship between exposure to green space and nature connection with mental health outcomes differed before and during the coronavirus pandemic.

In line with aim (i), independent *t*-tests revealed that pre-pandemic survey respondents had significantly higher well-being and lower depression, anxiety, and stress scores than survey respondents during the pandemic, with the pandemic mean for all variables being significantly worse than their respective normative values. However, the generalized linear modelling revealed that the survey time point did not significantly predict scores on any mental health outcome measures. Although evidence has consistently demonstrated the adverse impacts of the pandemic [7,8,9], it has also been identified that the pandemic had the most detrimental effects in younger adults, females and ethnic minority groups [12,13]. Our modelling techniques explored the difference between time points after accounting for variables already entered into the model, which included age, gender, nature connection and exposure to green space, thus potentially explaining the lack of effect.

In relation to aim (ii), the findings revealed that after accounting for age and gender, greater nature connection was associated with significantly lower depression and stress, and improved well-being. This is in line with previous research that has demonstrated that nature connection is positively associated with well-being, increased happiness, more purposeful, fulfilling, and meaningful lives [43], a lower likelihood of medication for depression, reduced mental distress, and higher life satisfaction [29,30,31]. Furthermore, the relationship between nature connection and mental well-being has been found in both the general population and those experiencing mental ill-heath [44,45,46], further demonstrating the potential importance of this construct. Whilst nature connection was not significantly associated with anxiety, previous research has proposed that the nature–health relationship may differ between mental health conditions [46,47]. Anxiety is more likely to be caused by things happening in the present moment, with different physiological effects from depression [48,49].

The percentage of green space in residential ward was not significantly associated with scores on any of the mental health outcomes after controlling for age and gender. Previous research has demonstrated that both exposure and access to green space is beneficial for mental health outcomes [18,19,20,21,22,23,24]. These contrasting findings could be a result of the variety of measures of green space ‘exposure’ utilised in the literature. In the current study, we used an estimate of green space presence in small defined areas [42]; however, the measure included only green spaces larger than 5 m^2^ and did not take into consideration whether the green space was accessible, of high quality or use. Other studies looked specifically at access rather than exposure to green space. For example, Hubbard et al. [21] reported that both physical access and visits to green space were associated with reduced psychological distress during the pandemic, whilst van den Berg et al. [50] found purposeful visits to green space pre-pandemic were associated with better mental health and higher vitality. Furthermore, it is suggested that nature contact may mediate the relationship between nature connectedness and health, as individuals who are more connected to nature are also more likely to visit green spaces and thus experience health benefits [29,51], potentially explaining the importance of nature connection over exposure to green space.

The final aim of the study (iii) was to explore whether the relationship between exposure to green space and nature connection with mental health outcomes differed before and during the coronavirus pandemic. Although Model 3 represented the best fit for the data, the interaction of the study time point with green space exposure and nature connection was not a significant predictor of any of the outcome measures. This finding suggests that the relationships between green space exposure and nature connection with mental health outcomes were consistent across the two time points. For depression only, the interaction of time point and nature connection was approaching significance. The Wald chi-squared statistic indicated a stronger relationship between nature connection and depression pre-COVID compared to during the coronavirus pandemic. This finding could suggest that during the pandemic, there were other unmeasured variables that played a more important role in mental health, leading to a reduction in the importance of nature connection. For example, social interaction, which was restricted during the pandemic, might have had a greater effect on depression scores during this time.

This study supports findings of previous research that indicated that nature connection is a significant component of the nature–health relationship. Nature-based interventions are already being used to support various populations, including people who have experienced trauma and people living with mental ill health [52,53,54]. To fully understand the role of nature connection for different populations, we recommend that nature-based interventions actively seek to increase nature connection through activities that promote meaningful connection with the natural environment [55,56]. Increasing nature connection could both improve health outcomes and increase cost effectiveness of interventions.

The current study has a number of limitations that need consideration. Whilst the data were collected from a large sample of the general population, neither the overall dataset nor sub-samples are representative of the UK general population, with a majority of participants being female, white British, and in full-time employment. The percentage of accessible green space was lower in the ‘during’ group and may have contributed to the lower well-being scores reported by this group. Further, demographic variables not controlled for in the models, such as ethnicity and employment status, may have affected mental health outcomes. Nature connection scores were also significantly greater than the UK average in both samples. This therefore limits generalisation of the findings, and the exploration of how demographic factors relate to mental health outcomes. The data were also collected from two different sub-groups, as opposed to representing repeated measurements in the same sample over time. Whilst these repeated comparisons would have enabled further exploration of changes in mental health outcomes over the course of the pandemic, generalised linear modelling techniques were employed to robustly explore factors associated with mental health outcomes. To build on the data in the current study and to further explore whether the relationship between nature connection, green space and mental health outcomes differs as a result of the pandemic, it would be beneficial to collect data in comparative samples. These additional data would support comparisons of relationships pre-, during, and post-pandemic and further elucidate the longer-term influence of the coronavirus pandemic on these relationships.

## 5. Conclusions

Overall, the findings of this study indicate that nature connection is significantly associated with decreased depression, anxiety, stress and increased well-being after accounting for the effects of age and gender. Whilst the pandemic resulted in worsened scores on all mental health variables, scores did not differ by study time point after accounting for age, gender and nature connection, further emphasising their importance. Strategies to improve mental health and reduce mental illness should consider the role of nature connection and the use of interventions that involve direct interaction with natural environments.

## Figures and Tables

**Table 1 ijerph-20-05083-t001:** Demographic information of participants.

		Pre-Pandemic *n* = 470 *n* (%) Mean (st. dev.)	During-Pandemic *n* = 407 *n* (%) Mean (st. dev.)	Total Sample *n* = 877 *n* (%) Mean (st. dev.)
Age (years) *		48.46 (14.42)	40.79 (14.32)	44.91 (14.87)
Gender *	Male	183 (38.9%)	102 (25.1%)	285 (32.5%)
	Female	286 (60.9%)	304 (74.7%)	590 (67.3%)
	Other	1 (0.2%)	1 (0.2%)	2 (0.2%)
Ethnicity	White	447 (96.8%)	388 (96.8%)	835 (96.8%)
	Asian	6 (1.2%)	5 (1.1%)	11 (1.3%)
	Mixed	6 (1.3%)	8 (1.9%)	14 (1.6%)
	Other	3 (0.6%)	0 (0.0%)	3 (0.3%)
Employment *	Full-time	197 (42%)	207 (51.0%)	404 (46.2%)
	Part-time	78 (16.6%)	73 (18.0%)	151 (17.3%)
	Self-employed	51 (10.9%)	32 (7.9%)	83 (9.5%)
	Unemployed/unable to work	14 (2.9%)	8 (1.9%)	22 (2.6%)
	Retired	92 (19.6%)	42 (10.3%)	134 (15.3%)
	Homemaker	6 (1.3%)	11 (2.7%)	17 (1.9%)
	Student	31 (6.6%)	24 (5.9%)	55 (6.3%)
	Other	0 (0.0%)	9 (2.2%)	9 (1%)
Relationship status	Single (legally)	132 (28.1%)	111 (27.3%)	243 (27.7%)
	Married	282 (60.0%)	268 (66%)	550 (62.8%)
	Divorced or separated	47 (10.0%)	22 (5.4%)	69 (7.9%)
	Widowed	9 (1.9%)	5 (1.2%)	14 (1.6%)
Percentage of green space in ward *		62.38 ± 26.37	56.30 ± 24.95	59.40 ± 25.84
Nature connection		70.46 ± 23.94 ^#1^	69.73 ± 23.67 ^#1^	70.13 ± 23.81 ^#1^

Note: 14 participants did not disclose their ethnicity, 2 participants did not disclose their employment status and 1 did not disclose their relationship status; * indicates significant difference between pre- and during-pandemic groups (*p* < 0.05); ^#1^ indicates significantly higher mean than pre-pandemic norm (*p* < 0.001).

**Table 2 ijerph-20-05083-t002:** Mean ± SD of depression, anxiety, stress and well-being variables pre- and during- the pandemic.

	Pre-Pandemic (*n* = 470)	During-Pandemic (*n* = 407)
Depression	7.51 ± 8.84 ^#1^	9.89 ± 8.40 *^,#1^
Anxiety	3.67 ± 5.94	5.62 ± 6.69 *^,#1^
Stress	9.29 ± 8.59	12.86 ± 8.65 *^,#1^
Well-Being	22.46 ± 3.96 ^#1^	21.70 ± 3.77 *^,#1,#2^

* Indicates significantly worse score during the pandemic (*p* < 0.05); ^#1^ indicates significantly worse score than pre-pandemic norm (*p* < 0.001); ^#2^ indicates significantly better score than pandemic norm (*p* < 0.001).

**Table 3 ijerph-20-05083-t003:** Model fit statistics for outcome variables.

	Model 1		Model 2		Model 3	
	AIC	Model Fit	AIC	Model Fit	AIC	Model Fit
Depression	5566.96	χ^2^ (2) = 35.44 *p* < 0.002	3901.59	χ^2^ (4) = 37.25 *p* < 0.001	3897.32	χ^2^ (7) = 47.52 *p* < 0.001
Anxiety	4372.30	χ^2^ (2) = 53.86 *p* < 0.001	3052.32	χ^2^ (4) = 35.69 *p* < 0.001	3050.20	χ^2^ (7) = 42.82 *p* < 0.001
Stress	5899.12	χ^2^ (2) = 45.12 *p* < 0.001	4168.28	χ^2^ (4) = 33.75 *p* < 0.001	4163.10	χ^2^ (7) = 44.94 *p* < 0.001
Well-Being	4835.52	χ^2^ (2) = 15.17 *p* < 0.001	3377.79	χ^2^ (4) = 38.89 *p* < 0.001	3374.55	χ^2^ (7) = 48.13 *p* < 0.001

Note: AIC: Akaike’s information criterion, where a lower value indicates improved model fit.

**Table 4 ijerph-20-05083-t004:** Parameter estimates for model predictors.

		Model 1			Model 2			Model 3		
		Beta	Wald Chi-Square	Sig	Beta	Wald Chi-Squared	Sig	Beta	Wald Chi-Squared	Sig
Depression	Age	−0.015	35.012	<0.001 *	−0.013	20.644	<0.001 *	−0.011	14.254	<0.001 *
	Gender	−0.080	1.126	0.289	−0.094	1.166	0.280	−0.114	1.673	0.196
	Nature connection	-	-	-	−0.007	18.070	<0.001 *	−0.011	16.157	<0.001 *
	Percentage green space	-	-	-	0.00	0.003	0.955	−0.001	0.089	0.766
	Time point	-	-	-	-	-	-	−0.364	1.647	0.199
	Time point *nature connection	-	-	-	-	-	-	0.006	3.629	0.057
	Time point *percentage green space	-	-	-	-	-	-	0.002	0.604	0.437
Anxiety	Age	−0.021	38.385	<0.001 *	−0.019	25.711	<0.001 *	−0.016	19.445	<0.001 *
	Gender	0.315	9.006	0.003 *	0.235	3.831	0.050	0.200	2.797	0.094
	Nature connection	-	-	-	−0.003	1.715	0.190	−0.002	0.256	0.613
	Percentage green space	-	-	-	−0.004	2.809	0.094	−0.005	2.754	0.097
	Time point	-	-	-	-	-	-	0.241	0.326	0.568
	Time point *nature connection	-	-	-	-	-	-	−0.002	0.232	0.630
	Time point *percentage green space	-	-	-	-	-	-	0.004	0.907	0.341
Stress	Age	−0.013	40.358	<0.001 *	−0.011	24.147	<0.001 *	−0.009	16.464	<0.001 *
	Gender	0.128	4.364	0.037 *	0.113	2.689	0.101	0.082	1.417	0.234
	Nature connection	-	-	-	−0.004	7.093	0.008 *	−0.004	4.286	0.038 *
	Percentage green space	-	-	-	−0.002	1.818	0.178	−0.002	0.934	0.334
	Time point	-	-	-	-	-	-	0.078	0.116	0.733
	Time point *nature connection	-	-	-	-	-	-	0.001	0.263	0.608
	Time point *percentage green space	-	-	-	-	-	-	0.001	0.230	0.632
Well-being	Age	0.035	14.065	<0.001 *	0.025	5.306	0.021 *	0.020	2.869	0.090
	Gender	0.178	0.382	0.537	0.042	0.016	0.901	0.067	0.040	0.842
	Nature connection	-	-	-	0.036	32.469	<0.001 *	0.047	30.260	<0.001 *
	Percentage green space	-	-	-	0.002	0.076	0.783	0.009	1.453	0.228
	Time point	-	-	-	-	-	-	1.823	3.116	0.078
	Time point *nature connection	-	-	-	-	-	-	−0.021	2.752	0.097
	Time point *percentage green space	-	-	-	-	-	-	−0.017	2.394	0.122

* Indicates a significant predictive effect of the variable on the outcome measure. Gender = 0 for males and 1 for females. Time point = 0 for pre-COVID and 1 for during COVID. Beta indicates the number of SDs the scores in the dependent variable would change if there were a one SD-unit change in the predictor.

## Data Availability

The data presented are available on request from the authors.

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
