# Peer review of "The Impact of the Coronavirus Pandemic on the Contribution of Local Green Space and Nature Connection to Mental Health"

_ijerph, 2023, doi:10.3390/ijerph20065083_

Round 1

Author Response

Thank you for your helpful comments. The attached file contains the responses to  each of the comments provided. 

Reviewer 2 Report

Introduction includes references to the research on the benefits of exposure to green space for mental health, but there is little reference to the research on relationships between both mental health and the coronavirus pandemic and the exposure to green space and mental health, which are more a subject of the article, so should be highlighted. Because of that the introduction is not complete and not all relevant references were cited, need to be complemented. Some are already mentioned in discussion.

Concerning the appropriation of the research sample, one thing raises the doubts. Namely, (223-225) "Participants who completed the survey during the pandemic had significantly worse scores on all wellbeing variables and a lower percentage of green space within the ward in which they lived". If there was a lower percentage of green space within the ward in which they lived during the pandemic, perhaps it was the reason for their worse mental health, not the pandemic. Both samples should have the same percentage of green space within the ward in which they lived.

Author Response

(The authors gave the same response as above.)

Round 2

Reviewer 1 Report

Please see reviewer 1 responses to author responses in blue.

Author Response

Thank you for your comments. Please find responses in the attached file. 
